# Impact of Persistent Multidrug-Resistant Gram-Negative Bacteremia on Clinical Outcome and Mortality

**DOI:** 10.3390/antibiotics12020313

**Published:** 2023-02-03

**Authors:** Shiori Kitaya, Hajime Kanamori, Yukio Katori, Koichi Tokuda

**Affiliations:** 1Department of Infectious Diseases, Internal Medicine, Tohoku University Graduate School of Medicine, Sendai 980-8574, Japan; 2Department of Otolaryngology, Head and Neck Surgery, Tohoku University Graduate School of Medicine, Sendai 980-8574, Japan

**Keywords:** persistent bacteremia, Enterobacterales, glucose-nonfermenting gram-negative rod, antimicrobial resistance, follow-up blood culture, clearance of persistent bacteremia

## Abstract

The clinical aspects of persistent bacteremia (PB) caused by gram-negative rods (GNRs) in terms of antimicrobial resistance (AMR) and PB clearance status are unclear. This secondary analysis of a retrospective cohort study investigated differences in PB caused by Enterobacterales and glucose non-fermentative GNRs (NF-GNRs) based on AMR and PB clearance. We retrospectively surveyed medical records at Tohoku University Hospital. Patients for whom blood cultures were performed between January 2012 and December 2021 were recruited. PB cases were grouped based on AMR and PB clearance; the characteristics of PB due to each bacterial pathogen were examined. The main outcome variable was mortality. The late (30–90-day) mortality rate was significantly higher in the multidrug-resistant (MDR) group than in the non-MDR group for Enterobacterales. However, no significant difference was noted in mortality rates between NF-GNRs with and without AMR. Mortality rates tended to be higher in the non-PB-clearance group than in the clearance group for both Enterobacterales and NF-GNRs. Since the mortality rate was higher in the MDR group in the case of Enterobacterales PB, more careful management is necessary for this condition. Follow-up blood cultures and confirming the clearance of PB are useful for improving the survival rate.

## 1. Introduction

Persistent bacteremia (PB) caused by gram-negative rods (GNRs) is associated with a higher mortality rate than non-PB [1]; thus, more careful management is required during its treatment. Our previous study found that GNR-PB accounted for 24.2% (100/414 cases) of all PB cases and that the mortality rate was higher in the PB non-clearance group than in the PB clearance group [2]. Several studies on the risk factors for GNR-PB have been reported, and endovascular devices, cardiac devices, hemodialysis, corticosteroids, epidural abscesses, septic thrombus, non-fermenter, and multidrug resistance, such as extended-spectrum β-lactamase or carbapenem resistance, have been identified as independent risk factors [3,4,5,6].

The spread of multidrug-resistant (MDR) gram-negative bacterial isolates, including extended-spectrum β-lactamases, carbapenem-resistant Enterobacterales, *Acinetobacter baumannii*, and *Pseudomonas aeruginosa*, are on the rise worldwide [7,8,9,10]. The most common mechanism for the development of multidrug resistance in gram-negative bacteria involved the horizontal transfer of plasmids carrying resistance genes, particularly carbapenemases [11]. The utilization of newly developed agents such as tigecycline, ceftolozane–tazobactam, and ceftazidime–avibactam as empirical therapy demonstrates the superior efficacy of these agents against MDR gram-negative bacteria and thus represents a hopeful alternative to extant therapeutics; however, the available treatment options remain inadequate [12]. Additionally, infections caused by MDR strains, including bacteremia, lead to increased morbidity and mortality and prolonged hospital stays and thus place a significant burden on the healthcare system [13,14].

Although several studies on GNR-PB have been published in recent years, there are no reports comparing the clinical characteristics of PB between Enterobacterales and glucose non-fermentative GNRs (NF-GNRs). Additionally, there are no reports comparing the clinical outcomes of PB due to Enterobacterales and NF-GNRs with antimicrobial resistance (AMR), or PB clearance. Therefore, in this study, we investigated for the first time the clinical outcomes and mortality rates associated with PB due to Enterobacterales and NF-GNRs, with a focus on AMR and PB clearance.

## 2. Results

### 2.1. Clinical Characteristics of PB between Enterobacterales and NF-GNRs and in Terms of AMR Status

The clinical characteristics of PB between Enterobacterales and NF-GNRs and in terms of AMR status are described in detail below and shown inTable 1. Regarding the differences in the clinical characteristics of PB due to Enterobacterales and NF-GNRs, catheter-related bloodstream infection (CRBSI) was significantly less common in the Enterobacterales group than in the NF-GNR group (odds ratio [OR] = 0.4, *p* = 0.026). The rate and duration of admission to the high care unit (HCU) were significantly higher and longer in the Enterobacterales group than in the NF-GNR group (*p* = 0.017 for both).

Regarding the differences in the characteristics of PB concerning AMR, intravascular device implantation rates were significantly higher in the resistant NF-GNR group than in the susceptive NF-GNR group (*p* = 0.025). C-reactive protein levels were significantly higher in the MDR Enterobacterales group than in the non-MDR Enterobacterales group (*p* = 0.027). As for mortality rates, the late (30–90-day) mortality rate was approximately 7.7 times higher in the MDR Enterobacterales group than in the non-MDR Enterobacterales group (OR = 7.7, *p* = 0.014). On the other hand, for NF-GNRs, there was no significant difference in the mortality rates observed between the susceptible and resistant NF-GNR groups.

### 2.2. Clinical Characteristics of Enterobacterales and NF-GNR-PB in Terms of PB Clearance

The details regarding patient’s clinical characteristics of Enterobacterales and NF-GNR-PB in terms of PB clearance are shown in Table 2. For Enterobacterales, the length of hospital and intensive care unit (ICU) stays were significantly shorter in the PB non-clearance group than in the PB clearance group (*p* = 0.004 and *p* = 0.017, respectively). The median patient age was significantly higher in the PB non-clearance group than in the PB clearance group (*p* = 0.046). The late (30–90-day) and 90-day mortality rates were approximately 12 times higher in the PB non-clearance group than in the PB clearance group (OR = 11.8 and *p* = 0.003 and OR = 12.1 and *p* = 0.001, respectively).

For NF-GNRs, there were significantly more cases of unknown infection focus in the PB non-clearance group than in the PB clearance group (OR = 21, *p* = 0.011). Further, mortality tended to be higher in the PB non-clearance group than in the PB clearance group; the intergroup difference was not significant.

Multivariate logistic regression analysis revealed that none of the variables differed significantly as independent risk factors for early (30-day), late (30–90-day), and 90-day mortality.

Figure 1 shows reasons for non-clearance of persistent gram-negative bacteremia. The most common reason for not confirming the clearance of Enterobacterales PB was that a follow-up blood culture (FUBC) was not performed owing to improved clinical symptoms, the rate of which was significantly higher than in the NF-GNR-PB group (*p* = 0.001). The next most common reasons for non-clearance of Enterobacterales PB were inappropriate antimicrobial drug use, unknown infection focus, and insufficient source control. On the other hand, in many cases of PB caused by NF-GNRs, the clearance of PB was not confirmed owing to the best supportive care policy or death from non-infectious diseases, such as extended burn injury, cerebral infarction, acute renal failure, liver failure, and pancreatic cancer, the rate of which was significantly higher than in the Enterobacterales PB group (*p* = 0.029). The next most common reasons for the lack of clearance in PB caused by NF-GNR were catheter colonization and the absence of a clear explanation.

## 3. Discussion

### 3.1. Differences in Clinical Characteristics of PB between Enterobacterales and NF-GNR and in Terms of AMR

#### 3.1.1. Enterobacterales vs. NF-GNRs

Recent reports have shown an increase in the rate of gram-negative bacteremia in CRBSI [15,16,17]. The incidence of CRBSI was significantly lower in the Enterobacterales PB group than in the NF-GNR-PB group in this study. Concerning the factors associated with CRBSI, the NF-GNR-PB group had a higher rate of intravascular device insertion and a lower rate of removal than did the Enterobacterales PB group. The risk for CRBSI is increased by longer dwelling times of intravascular devices [18], and longer durations may contribute to increased CRBSI in the NF-GNR-PB group. Focusing on the characteristics of patients in the NF-GNR-PB group, those with an endovascular device were more likely to be admitted to the ICU and intubated than those without intravascular devices (10/20 cases [50%] vs. 2/8 cases [25%], and 11/20 cases [55%] vs. zero, respectively). Both ICU patients and intubated patients were considered to be in poor general condition and often required multidisciplinary treatment. Therefore, catheters are frequently implanted for administering drugs or nutrients, but it was considered difficult to remove or change intravascular devices as indicated.

The ICU and HCU admission rates tended to be higher in the Enterobacterales PB group than in the NF-GNR-PB group, and the intergroup difference in the latter was significant. In our hospital, a high proportion of the patients admitted to the ICU and HCU are postoperative abdominal and cardiovascular surgery patients. Cardiovascular surgery patients are frequently intubated and ventilated, and suctioning procedures are frequently performed to prevent and treat ventilator-associated pneumonia [19]. In addition, postoperative patients with gastrointestinal or hepatobiliary-pancreatic diseases are often implanted with abdominal drains or have stoma formation and frequently receive wound care [20,21]. The dispersal of sputum-containing enteric bacteria due to these suctioning procedures and the transmission of bacteria during wound care may contribute to the increased rates of healthcare-associated infections and PB due to enteric bacteria in ICUs and HCUs.

#### 3.1.2. AMR Status

The resistant NF-GNR group had a higher rate of intravascular device insertion than did the susceptive NF-GNR group. PB caused by antimicrobial-resistant GNR has been associated with poor clinical outcomes, such as a long hospital stay and treatment failure [22,23]. Therefore, patients with PB due to antimicrobial-resistant GNRs are expected to receive frequent and prolonged intravascular indwelling catheters for therapeutic purposes, including antimicrobial administration. Regarding CRBSI, the number of patients with CRBSI among those with endovascular indwelling devices was almost equal in the susceptible and resistant strain groups (5/10 cases, 50% vs. 5/10 cases, 50%; half of the cases of CRBSI in the resistant NF-GNR group were attributable to the use of a peripheral infusion route). Of these, nine cases (90%) in the susceptive group and four (80%) in the resistant group had adequate source control, such as catheter removal and replacement, and the corresponding numbers of death in these groups were three (33.3%) and zero. Therefore, these findings suggest that despite the limitation of a small number of cases, it is possible that in cases of CRBSI with NF-GNRs, even if the causative organism has AMR, the mortality rate can be kept almost the same as that in cases of infection with non-resistant pathogens if appropriate source control is performed.

MDR bacterial infections, including those with MDR Enterobacterales such as *Enterobacter cloacae*, tend to be refractory due to the limited treatment options involving antimicrobial agents [24]. Additionally, the treatment of MDR bacterial infections is complicated by the need for customizing treatment based on individual patient history, source of infection, comorbid conditions, and underlying bacterial resistance mechanisms [25,26,27]. Furthermore, a former study showed that patients with bloodstream infections (BSIs) with MDR bacteria have a higher mortality rate than those with BSIs with non-MDR bacteria [28]. Therefore, appropriate source control as well as antimicrobial therapy is considered more important in the treatment of BSIs caused by MDR bacteria [29]. In this study, the number of cases in which appropriate source control was performed was smaller in the MDR group than in the non-MDR group (11 cases, 45.8% vs. 25 cases, 52.1%), which may also have contributed to the increase in the late (30–90-day) mortality rate in the MDR group.

### 3.2. Differences in Clinical Characteristics of Enterobacterales and NF-GNR-PB in Terms of PB Clearance

A previous study has reported higher mortality rates for GNR-PB than for non-GNR-PB [1]. Our previous study found that the mortality rate was higher in the GNR-PB non-clearance group than in the GNR-PB clearance group [2]. In this study, we found that regarding GNRs, the PB non-clearance group tended to have higher mortality rates than did the PB clearance group for both Enterobacterales and NF-GNRs. In Enterobacterales bacteremia, acute pancreatitis, abdominal surgery, antacid use, patient age, acute physiology, chronic health evaluation II score, tracheal intubation or incision, and positivity of extended-spectrum β-lactamase organisms were reported as prognostic factors [30]. On the other hand, in NF-GNR, bacteremia, an indwelling central venous catheter (CVC), and steroid use were predisposing factors; cirrhosis, hematologic malignancies, pneumonia, septic shock, and ICU infection were reported to be fatal independent risk factors [31]. Although the clinical characteristics differ between the Enterobacterales and NF-GNR-PB groups, the present study shows that confirming the clearance of PB by FUBC contributes to increased survival in both of them.

For PB cases, appropriate source control, such as incision drainage or removal of catheters, is recommended by the Department of Infectious Diseases when the cause of infection is clear. Furthermore, focus identification using imaging modalities, such as computed tomography or gallium scintigraphy, is recommended when the focus of the infection is unknown. Even in cases of GNR-PB, the Department of Infectious Diseases encourages physicians in the main department to aggressively perform FUBC to confirm the clearance of PB, especially in severe cases. In this study, the most common reason for failure to achieve the clearance of PB in the Enterobacterales PB group was the failure to perform FUBC owing to the improvement of the general condition. Fortunately, there were no deaths among patients with Enterobacterales PB for whom FUBC was not performed at the discretion of the main department, owing to improvement in their general condition. However, the number of those cases was limited, and we found that the mortality rate was higher in patients without clearance of PB than in patients with clearance of both Enterobacterales and NF-GNR-PB.

Concerning infection focus, a previous review stated that aggressive source control is important in addition to appropriate antimicrobial therapy to improve clinical outcomes in patients with GNR-PB [6]. However, when the infection focus is unknown, proper source control is impossible. It is thought that there are cases in which a clear infection focus could not be identified even after the detailed examination, or in which a detailed examination itself is not performed due to the best supportive care policy or for other reasons. It is difficult to perform appropriate source control in those cases, and it is conceivable that this may lead to non-clearance of PB.

## 4. Materials and Methods

### 4.1. Study Design and Setting

We performed a secondary analysis based on a retrospective cohort study conducted at Tohoku University Hospital, Sendai, Miyagi, Japan. Electronic clinical charts and hospital records were reviewed between January 2012 and December 2021 to collect study variables [2]. Briefly, the data for each bloodstream isolate were collected from the computerized records of the Department of Laboratory Medicine, and the following anamnestic and clinical data were obtained from the medical records and the database of the Department of Infectious Diseases: sex, age, ethnicity, comorbidities, body mass index, and body temperature; blood test results, including serum white blood cell and neutrophil counts and C-reactive protein level; presence of intravascular devices, i.e., a central line such as a CVC, a peripherally inserted central catheter, tunneled CVC, and implanted central venous port, and removal of such devices; a history of cardiovascular surgery including valve replacement, vascular graft replacement, and ventricular assist device and cardiac device implantation; use of extracorporeal membrane oxygenation, continuous hemodiafiltration, and mechanical ventilation; interval between initial BC and FUBC; duration of bacteremia; site of infections; duration of hospitalization; whether time was spent in an ICU, HCU, or coronary care unit between the initial BC and the last FUBC; antibiotic use; performance of the source control; and mortality, recorded as early (30-day), late (30–90-day), and 90-day mortality. All patients diagnosed with BSI, defined as one or more positive BCs obtained for ruling out an infection, were eligible for inclusion. The data on microorganisms were extracted from the database of the infection department. This study was approved by the Human Ethical and Clinical Trial Committee of Tohoku University Hospital (approval: 2019-1-270). The requirement of patient consent was waived because of the retrospective nature of the study.

### 4.2. Definitions and Outcomes

The definition of FUBC, determination of the source of PB, duration of bacteremia, contamination, PB clearance, source control, neutropenia, immunosuppression, BC collection, adequacy of antimicrobial therapy, and patient selection algorithm were in accordance with our previous report [2]. The primary outcome variable in this study was mortality: early (30-day), late (30–90-day), and 90-day mortality within each day after the initial BC.

The minimum inhibitory concentrations and clinical breakpoints of each bacterium were evaluated according to the Clinical Laboratory Standards Institute guidelines [32]. MDR Enterobacterales was defined as a group of organisms with acquired non-susceptibility to at least one agent in three or more antimicrobial categories [33]. Resistant NF-GNR strains were defined as those resistant to one or more of the following antibacterial categories: aminoglycosides, quinolones, and carbapenems.

### 4.3. Statistical Analysis

Results are expressed as median values with 95% confidence intervals or as proportions of the total number of patients or isolates. Regarding comparisons between two groups, the Mann–Whitney U test was used to compare the averages of continuous variables, and Fisher’s exact test was used to compare the proportions of categorical variables. We performed a multivariate logistic regression analysis to identify risk factors for early (30 days), late (30–90-day), and 90-day mortality. All variables with *p*-values of less than 0.1 in the univariable analysis were included in the multivariate analysis. The analysis was performed using JMP pro 16 statistical analysis software (SAS institute, 2021). Differences were considered significant at a corrected *p*-value of <0.05.

## 5. Conclusions

To our knowledge, this is the first study investigating the characteristics of Enterobacterales and NF-GNR-PB from the viewpoint of both AMR and PB clearance status. The limitation of our study is that this is a single-center study and the number of patients with NF-GNR-PB was small; so, it is not clear whether generalization is possible. However, this is a long-term, decadal GNR-PB study, and the findings are considered clinically significant. The main conclusions of this study are as follows: (1) PB caused by MDR Enterobacterales is associated with significantly higher late (30–90-day) mortality than PB caused by non-MDR Enterobacterales, and (2) the PB non-clearance group had significantly higher mortality than did the PB clearance group for both Enterobacterales and NF-GNR-PB. Therefore, confirming the clearance of both Enterobacterales and NF-GNR-PB is important to improve survival rates.

## Figures and Tables

**Figure 1 antibiotics-12-00313-f001:**
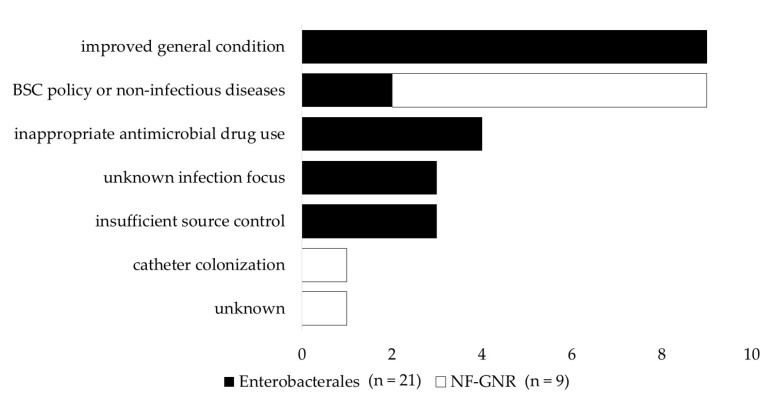
Reasons for non-clearance of persistent gram-negative bacteremia. An improved general condition means that a negative blood culture was not confirmed owing to an improved general condition. The underlying diseases among patients in line with the best supportive care policy were gastric cancer and graft-versus-host disease. A non-infectious disease is a case in which the clearance of persistent bacteremia could not be confirmed due to the death of the patient from a disease other than an infectious disease, extensive burn injury, cerebral infarction, acute renal failure, liver failure, and pancreatic cancer. Antimicrobial therapy was considered inappropriate when at least one of the following conditions was met: administration of ineffective antimicrobial agents, i.e., agents that did not effectively treat infections with organisms identified in the blood culture; continuation of the initial antimicrobial agents even though the result of the susceptibility test was known and de-escalation was possible; and administration of antibiotic therapy for a shorter time than the current medical standards. NF-GNR, glucose non-fermentative gram-negative rod.

**Table 1 antibiotics-12-00313-t001:** Clinical characteristics of persistent bacteremia between Enterobacterales and glucose non-fermentative gram-negative rods and in terms of antimicrobial resistance status.

	Enterobacterales (n = 72)	NF-GNR (n = 28)	Odds Ratio [95% CI]	*p*-Value	MDR Enterobacterales (n = 24)	Non-MDR Enterobacterales (n = 48)	Odds Ratio [95% CI]	*p*-Value	Resistant NF-GNR (n = 10)	Susceptive NF-GNR (n = 18)	Odds Ratio [95% CI]	*p*-Value
**Demography**												
Sex (male, %)	51 (70.8)	16 (57.1)	1.8 [0.7, 4.5]		16 (66.7)	35 (72.9)	0.7 [0.3, 2.2]		7 (70.0)	9 (50.0)	2.3 [0.5, 12]	
Age, years, median (IQR)	59.5 (48.8–73.8)	71.5 (46.8–78.3)			69.0 (59.5–78.5)	57.0 (51.0–63.0)			62.0 (47.3–78.3)	72.5 (48.5–77.8)		
Ethnicity (Asian, %)	72 (100)	28 (100)	0		24 (100)	48 (100)	0		10 (100)	18 (100)	0	
**Comorbidities**												
Diabetes mellitus	13 (18.1)	2 (7.1)	2.9 [0.6, 13.6]		6 (25.0)	7 (14.6)	2 [0.6, 6.6]		0 (0)	2 (11.1)	0	
ESDR on hemodialysis	4 (5.6)	2 (7.1)	0.8 [0.1, 4.4]		1 (4.2)	3 (6.3)	0.7 [0.1, 6.6]		1 (10.0)	1 (5.6)	1.9 [0.1, 33.9]	
Liver cirrhosis	4 (5.6)	1 (3.6)	1.6 [0.2, 14.9]		0 (0)	4 (8.3)	0		0 (0)	1 (5.6)	0	
Solid malignancy	21 (29.2)	8 (28.6)	1 [0.4, 2.7]		7 (29.2)	14 (29.2)	1.0 [0.3, 2.9]		2 (20.0)	6 (33.3)	0.5 [0.1, 3.1]	
Hematologic malignancy	4 (5.6)	2 (7.1)	0.8 [0.1, 4.4]		1 (4.2)	3 (6.3)	0.7 [0.1, 6.6]		2 (20.0)	0 (0)	–	
Neutropenia	1 (1.4)	3 (10.7)	0.1 [0, 1.2]		0 (0)	1 (2.1)	0		2 (20.0)	1 (5.6)	4.3 [0.3, 54.1]	
Immunosuppression	8 (11.1)	6 (21.4)	0.5 [0.1, 1.5]		4 (16.7)	5 (10.4)	1.7 [0.4, 7.1]		4 (40.0)	3 (16.7)	3.3 [0.6, 19.6]	
**Vital signs**												
BMI, kg/m², median (IQR)	21.2 (18.9–24.1)	23.1 (19.0–26.0)			22.9 (18.9–25.9)	20.6 (19.2–23.1)			22.0 (18.8–24.6)	23.1 (19.8–28.7)		
Body temperature, °C, median (IQR)	38.7 (37.8–39.3) (n = 68)	38.7 (38.0–39.2)			39.0 (38.2–39.3)	38.0 (37.6–39.3) (n = 44)			38.6 (37.6–39.3)	38.7 (38.1–39.2)		
**Laboratory markers**												
White blood cell count, 10^9^/L, median (IQR)	9700.0 (6125.0–13,450.0)	8650.0 (5075.0–14,225.0)			10,250.0 (7850.0–13,175.0)	9100.0 (5700.0–13,675.0)			8650.0 (4950.0–10,075.0)	9050.0 (5225.0–15,275.0)		
Neutrophil count, 10^9^/L, median (IQR)	8590.0 (5310.0–12,675.0)	7790.0 (3830.0–13,105.0)			9600.0 (6947.5–12,472.5)	8090.0 (4565.0–12,807.5)			7790.0 (4100.0–9372.5)	7870.0 (3575.0–14,475.0)		
C-reactive protein, mg/dL, median (IQR)	9.0 (4.1–15.0)	8.2 (3.4–13.0)			11.5 (7.0–21.5)	7.8 (3.8–12.9)		0.027	8.3 (4.1–15.3)	8.2 (3.4–11.1)		
**Devices**												
Intravascular device	46 (63.9)	20 (71.4)	0.7 [0.3, 1.8]		17 (70.8)	29 (60.4)	1.6 [0.6, 4.6]		10 (100)	10 (55.6)	–	0.025
Intravascular device removal	38 (82.6)	14 (70.0)	2 [0.6, 6.9]		16 (94.1)	22 (75.9)	5.1 [0.6, 45.6]		6 (60.0)	8 (80.0)	2.7 [0.4, 19.7]	
Cardiovascular surgery	25 (34.7)	6 (21.4)	2 [0.7, 5.4]		7 (29.2)	18 (37.5)	0.7 [0.2, 2]		2 (20.0)	4 (22.2)	0.9 [0.1, 5.9]	
ECMO	1 (1.4)	2 (7.1)	0.2 [0, 2.1]		1 (4.2)	0 (0)	–		0 (0)	2 (11.1)	0	
Continuous hemodiafiltration	17 (23.6)	7 (25.0)	0.9 [0.3, 2.6]		7 (29.2)	10 (20.8)	1.6 [0.5, 4.8]		2 (20.0)	5 (27.8)	0.7 [0.1, 4.2]	
Mechanical ventilation	32 (44.4)	11 (39.3)	1.2 [0.5, 3]		14 (58.3)	18 (37.5)	2.3 [0.9, 6.3]		6 (60.0)	5 (27.8)	3.9 [0.8, 20]	
**Status of persistent bacteremia**												
The period until FUBC is carried out, median (IQR)	3.0 (1.0–4.0)	2.0 (2.0–4.0)			3.0 (2.0–5.0)	3.0 (1.0–4.0)			2.0 (2.0–2.5)	2.5 (1.3–4.8)		
Duration of bacteremia, median (IQR)	4.0 (2.0–7.0)	3.0 (2.0–6.0)			3.0 (2.8–6.3)	4.0 (2.0–7.3)			2.5 (2.0–5.3)	4.0 (2.0–6.0)		
**Site of infection**												
CRBSI	22 (25.9)	15 (48.4)	0.4 [0.2, 0.9]	0.026	4 (14.3)	18 (31.6)	0.3 [0.1, 1.1]		5 (41.7)	10 (52.6)	0.6 [0.1, 2.8]	
Infectious endocarditis	2 (2.4)	0 (0)	–		0 (0)	2 (3.5)	0		0 (0)	0 (0)	0	
Septic embolism	2 (2.4)	0 (0)	–		0 (0)	2 (3.5)	0		0 (0)	0 (0)	0	
Endovascular devices infections	3 (3.5)	2 (6.5)	0.5 [0.1, 3.3]		1 (3.6)	2 (3.5)	2 [0.1, 34.2]		1 (8.3)	1 (5.3)	1.6 [0.1, 28.9]	
Thrombophlebitis	6 (7.1)	4 (12.9)	0.5 [0.1, 2]		2 (7.1)	4 (7.0)	1 [0.2, 5.9]		1 (8.3)	3 (15.8)	0.5 [0, 5.3]	
Pyogenic spondylitis	1 (1.2)	0 (0)	–		1 (3.6)	0 (0)	–		0 (0)	0 (0)	0	
Abscess	5 (5.9)	1 (3.2)	1.9 [0.2, 16.7]		1 (3.6)	4 (7.0)	0.5 [0.1, 4.5]		0 (0)	1 (5.3)	0	
Pneumonia	1 (1.2)	1 (3.2)	0.4 [0, 5.9]		0 (0)	1 (1.8)	0		1 (8.3)	0 (0)	–	
Intra-abdominal infections	4 (4.7)	0 (0)	–		2 (7.1)	2 (3.5)	2.1 [0.3, 15.8]		0 (0)	0 (0)	0	
Urinary tract infections	9 (10.6)	1 (3.2)	3.6 [0.4, 29.3]		5 (17.9)	4 (7.0)	2.9 [0.7, 12]		0 (0)	1 (5.3)	0	
Biliary tract infections	2 (2.4)	0 (0)	–		0 (0)	2 (3.5)	0		0 (0)	0 (0)	0	
Skin and soft tissue infections	1 (1.2)	0 (0)	–		0 (0)	1 (1.8)	0		0 (0)	0 (0)	0	
Surgical site infection	5 (5.9)	2 (6.5)	0.9 [0.2, 4.9]		3 (10.7)	2 (3.5)	3.3 [0.5, 21.2]		1 (8.3)	1 (5.3)	1.6 [0.1, 28.9]	
Sinusitis	1 (1.2)	0 (0)	–		0 (0)	1 (1.8)	0		0 (0)	0 (0)	0	
Mediastinitis	1 (1.2)	0 (0)	–		1 (3.6)	0 (0)	–		0 (0)	0 (0)	0	
Unknown	20 (23.5)	5 (16.1)	1.6 [0.5, 4.7]		8 (28.6)	12 (21.1)	1.5 [0.5, 4.4]		3 (25.0)	2 (10.5)	2.8 [0.4, 20.2]	
**Hospital stays**												
Duration of hospitalization, days, median (IQR)	69.0 (31.8–138.5)	102.5 (52.5–153.3)			59.5 (41.0–129.8)	75.5 (25.8–138.5)			141.5 (83.3–192.0)	93.0 (39.0–133.5)		
Presence of ICU	45 (62.5)	12 (42.9)	2.2 [0.9, 5.4]		17 (70.8)	28 (58.3)	1.7 [0.6, 5.0]		5 (50.0)	7 (38.9)	1.6 [0.3, 7.5]	
Duration of ICU stay, days, median (IQR)	8.0 (0–44.0)	0 (0–59.5)			9.0 (0–52.3)	7.0 (0–39.3)			9 (0–88.5)	0 (0–43.8)		
Presence of HCU	13 (18.1)	0 (0)	–	0.017	4 (16.7)	9 (18.8)	0.9 [0.2, 3.2]		0 (0)	0 (0)	0	
Duration of HCU stay, days, median (IQR)	0 (0–0)	0 (0–0)		0.017	0 (0–0)	0 (0–0)			0 (0–0)	0 (0–0)		
Presence of CCU	2 (2.8)	1 (3.6)	0.8 [0.1, 8.9]		0 (0)	2 (4.2)	0		0 (0)	1 (5.6)	0	
Duration of CCU stay, days, median (IQR)	0 (0–0)	0 (0–0)			0 (0–0)	0 (0–0)			0 (0–0)	0 (0–0)		
**Intervention**												
The use of antibiotics (Appropriate)	62 (86.1)	27 (96.4)	0.3 [0, 2.2]		22 (91.7)	41 (85.4)	1.9 [0.4, 9.8]		10 (100)	17 (94.4)	–	
Source control	36 (50.0)	16 (57.1)	0.8 [0.3, 1.8]		11 (45.8)	25 (52.1)	0.8 [0.3, 2.1]		5 (50.0)	11 (61.1)	0.6 [0.1, 3]	
**Mortality**												
Early (30-day) mortality	3 (4.2)	4 (14.3)	0.3 [0.1, 1.3]		0 (0)	3 (6.3)	0		1 (10.0)	3 (16.7)	0.6 [0, 6.2]	
Late (30–90-day) mortality	8 (11.1)	2 (7.1)	1.6 [0.3, 8.2]		6 (25.0)	2 (4.2)	7.7 [1.4, 41.6]	0.014	1 (10.0)	1 (5.6)	1.9 [0.1, 33.9]	
90-day mortality	11 (15.3)	6 (21.4)	0.7 [0.2, 2]		6 (25.0)	5 (10.4)	2.9 [0.8, 10.6]		2 (20.0)	4 (22.2)	0.9 [0.1, 5.9]	

Data are presented as number (%) unless indicated otherwise. In the table, *p*-values are listed only for items that show significant differences. The blood test was performed on the same day as the blood culture collection. Immunosuppression was considered in neutropenia, hematopoietic stem-cell transplantation, solid organ transplantation, and corticosteroid therapy (prednisone 16 mg per day for 15 days). Cardiovascular surgery includes valve replacement, vascular graft replacement, ventricular assist device, and cardiac device implantation. BMI, body mass index; CCU, coronary care unit; CI, confidence interval; CRBSI, catheter-related bloodstream infection; ECMO, extracorporeal membrane oxygenation; ESDR, end-stage renal disease; FUBC, follow-up blood culture; HCU, high care unit; ICU, intensive care unit; IQR, interquartile range; MDR, multidrug-resistant; NF-GNR, glucose non-fermentative gram-negative rod.

**Table 2 antibiotics-12-00313-t002:** Clinical characteristics of Enterobacterales and glucose non-fermentative gram-negative rod persistent bacteremia in terms of persistent bacteremia clearance.

	Enterobacterales PB Non-Clearance (n = 19)	Enterobacterales PB Clearance (n = 53)	Odds Ratio [95% CI]	*p*-Value	NF-GNR-PB Non-Clearance (n = 9)	NF-GNR-PB Clearance (n = 19)	Odds Ratio [95% CI]	*p*-Value
Demography								
Sex (male, %)	13 (68.4)	38 (71.7)	0.9 [0.3, 2.7]		5 (55.6)	11 (57.9)	0.91 [0.18, 4.50]	
Age, years, median (IQR)	73.0 (68.5–76.0)	59.5 (48.8–73.8)		0.046	72.0 (51.0–76.0)	71.0 (46.5–78.5)		
Ethnicity (Asian, %)	19 (100)	53 (100)	0		9 (100)	19 (100)	0	
**Comorbidities**								
Diabetes mellitus	5 (26.3)	8 (15.1)	2 [0.6, 7.1]		0 (0)	2 (10.5)	0	
ESDR on hemodialysis	2 (10.5)	2 (3.8)	3 [0.4, 23]		0 (0)	2 (10.5)	0	
Liver cirrhosis	2 (10.5)	2 (3.8)	3 [0.4, 23]		0 (0)	1 (5.3)	0	
Solid malignancy	9 (47.4)	12 (22.6)	3 [1, 9.3]		2 (22.2)	6 (31.6)	0.6 [0.1, 3.9]	
Hematologic malignancy	0 (0)	4 (7.5)	0		2 (22.2)	0 (0)	–	
Neutropenia	1 (5.3)	0 (0)	–		1 (11.1)	2 (10.5)	1.1 [0.1, 13.5]	
Immunosuppression	3 (15.8)	6 (11.3)	1.5 [0.3, 6.6]		4 (44.4)	3 (15.8)	4.3 [0.7, 25.9]	
**Vital signs**								
BMI, kg/m², median (IQR)	20.0 (18.4–22.9)	21.6 (19.7–25.1)			23.7 (19.1–27.4)	22.8 (18.8–25.2)		
Body temperature, °C, median (IQR)	38.8 (38.1–39.0) (n = 18)	38.3 (37.7–39.5) (n = 50)			38.8 (38.0–39.3)	38.6 (38.0–39.1)		
**Laboratory markers**								
White blood cell count, 10^9^/L, median (IQR)	10,100.0 (6750.0–14,500.0)	9300.0 (5900.0–13,100.0)			11,900.0 (4900.0–15,900.0)	8500.0 (5350.0–11,950.0)		
Neutrophil count, 10^9^/L, median (IQR)	9410.0 (5995.0–13,195.0)	7600.0 (5100.0–12,390.0)			9760.0 (4300.0–15,580.0)	7740.0 (3680.0–10,145.0)		
C-reactive protein, mg/dL, median (IQR)	4.7 (2.5–12.2)	9.4 (5.0–17.0)			14.2 (6.2–19.2)	7.2 (3.1–10.1)		
**Devices**								
Intravascular device	10 (52.6)	36 (67.9)	0.5 [0.2, 1.5]		8 (88.9)	12 (63.2)	4.7 [0.5, 45.4]	
Intravascular device removal	6 (60.0)	32 (88.9)	0.2 [0, 1]		6 (75.0)	8 (66.7)	1.5 [0.2, 11.1]	
Cardiovascular surgery	6 (31.6)	19 (35.8)	0.8 [0.3, 2.5]		1 (11.1)	5 (26.3)	0.4 [0, 3.5]	
ECMO	0 (0)	1 (1.9)	–		1 (11.1)	1 (5.3)	2.3 [0.1, 40.7]	
Continuous hemodiafiltration	2 (10.5)	15 (28.3)	0.3 [0.1, 1.5]		3 (33.3)	4 (21.1)	1.9 [0.3, 11]	
Mechanical ventilation	5 (26.3)	27 (50.9)	0.3 [0.1, 1.1]		4 (44.4)	7 (36.8)	1.4 [0.3, 6.9]	
**Status of persistent bacteremia**								
The period until FUBC is carried out, median (IQR)	3.0 (1.0–4.5)	3.0 (1.0–4.0)			2.0 (2.0–3.0)	3.0 (2.0–4.5)		
Duration of bacteremia, median (IQR)	4.0 (1.5–8.0)	4.0 (2.0–7.0)			4.0 (2.0–8.0)	3.0 (2.0–5.5)		
**Site of infection**								
CRBSI	6 (27.3)	16 (25.4)	1.1 [0.4, 3.3]		4 (44.4)	11 (50.0)	1 [0.2, 5]	
Infectious endocarditis	0 (0)	2 (3.2)	–		0 (0)	0 (0)	0	
Septic embolism	0 (0)	2 (3.2)	–		0 (0)	0 (0)	0	
Endovascular devices infections	0 (0)	3 (4.8)	–		0 (0)	2 (9.1)	–	
Thrombophlebitis	1 (4.5)	5 (7.9)	0.6 [0.1, 5]		0 (0)	4 (18.2)	–	
Pyogenic spondylitis	0 (0)	1 (1.6)	–		0 (0)	0 (0)	0	
Abscess	0 (0)	5 (7.9)	–		0 (0)	1 (4.5)	–	
Pneumonia	0 (0)	1 (1.6)	–		0 (0)	1 (4.5)	–	
Intra-abdominal infections	1 (4.5)	3 (4.8)	1 [0.1, 9.7]		0 (0)	0 (0)	0	
Urinary tract infections	4 (18.2)	5 (7.9)	2.6 [0.6, 10.6]		0 (0)	1 (4.5)	–	
Biliary tract infections	1 (4.5)	1 (1.6)	3 [0.2, 49.3]		0 (0)	0 (0)	0	
Skin and soft tissue infections	1 (4.5)	0 (0)	0		0 (0)	0 (0)	0	
Surgical site infection	2 (9.1)	3 (4.8)	2 [0.3, 12.8]		1 (11.1)	1 (4.5)	3 [0.2, 54.6]	
Sinusitis	1 (4.5)	0 (0)	0		0 (0)	0 (0)	0	
Mediastinitis	0 (0)	1 (1.6)	–		0 (0)	0 (0)	0	
Unknown	5 (22.7)	15 (23.8)	0.9 [0.3, 3]		4 (44.4)	1 (4.5)	21 [1.8, 240.5]	0.011
**Hospital stays**								
Duration of hospitalization, days, median (IQR)	45.0 (21.0–67.5)	95.0 (42.0–178.0)		0.004	76.0 (30.0–134.0)	105.5 (59.0–154.5)		
Presence of ICU	9 (47.4)	36 (67.9)	0.4 [0.1, 1.2]		3 (33.3)	9 (47.4)	0.6 [0.1, 2.9]	
Duration of ICU stay, days, median (IQR)	0 (0–12.0)	12.0 (0–54.0)		0.017	0 (0–33.0)	0 (0–75.0)		
Presence of HCU	0 (0)	8 (15.1)	–		0 (0)	0 (0)	0	
Duration of HCU stay, days, median (IQR)	0 (0–0.5)	0 (0–0)			0 (0–0)	0 (0–0)		
Presence of CCU	0 (0)	2 (3.8)	0		0 (0)	1 (5.3)	0	
Duration of CCU stay, days, median (IQR)	0 (0–0)	0 (0–0)			0 (0–0)	0 (0–0)		
**Intervention**								
The use of antibiotics (Appropriate)	16 (84.2)	47 (88.7)	0.7 [0.2, 3]		9 (100)	18 (94.7)	–	
Source control	7 (36.8)	29 (54.7)	0.5 [0.2, 1.4]		4 (44.4)	12 (63.2)	0.5 [0.1, 2.3]	
**Mortality**								
Early (30-day) mortality	2 (10.5)	1 (1.9)	6.1 [0.5, 71.8]		3 (33.3)	1 (5.3)	9 [0.8, 103.7]	
Late (30–90-day) mortality	6 (31.6)	2 (3.8)	11.8 [2.1, 65.2]	0.003	1 (11.1)	1 (5.3)	2.3 [0.1, 40.7]	
90-day mortality	8 (42.1)	3 (5.7)	12.1 [2.8, 53.2]	0.001	4 (44.4)	2 (10.5)	6.8 [0.9, 48.7]	

Data are presented as number (%) unless indicated otherwise. In the table, *p*-values are listed only for items that show significant differences. The blood test was performed on the same day as the blood culture collection. The blood test was performed on the same day as the blood culture collection. Immunosuppression was considered in neutropenia, hematopoietic stem-cell transplantation, solid organ transplantation, and corticosteroid therapy (prednisone 16 mg per day for 15 days). Cardiovascular surgery includes valve replacement, vascular graft replacement, ventricular assist device, and cardiac device implantation. BMI, body mass index; CCU, coronary care unit; CI, confidence interval; CRBSI, catheter-related bloodstream infection; ECMO, extracorporeal membrane oxygenation; ESDR, end-stage renal disease; FUBC, follow-up blood culture; HCU, high care unit; ICU, intensive care unit; IQR, interquartile range; MDR, multidrug-resistant; NF-GNR, glucose non-fermentative gram-negative rod; PB, persistent bacteremia.

## Data Availability

The datasets created and analyzed during the current study are not publicly available due to the fact that they contain a great deal of detailed patient information. The dataset is owned by the Department of Infectious Diseases, Internal Medicine, Tohoku University Graduate School.

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
