# Peer review of "Impact of Persistent Multidrug-Resistant Gram-Negative Bacteremia on Clinical Outcome and Mortality"

_antibiotics, 2023, doi:10.3390/antibiotics12020313_

Round 1

Reviewer 1 Report

Comments to the authors:

1. It would greatly help the reader if there were a table that included the resistance mechanism and the available treatment in the introduction section.

2. A few sentences on the epidemiology of multidrug resistance would benefit the manuscript, and general methods for antibiotic resistance would be valuable.

3. Clinical characteristics and outcome of Enterococcus: authors should expand on this a bit.

4. Can authors include the hospitalization time, if available, in Table 1?

5.  Authors should consider the association of the variables with mortality via the univariate and multivariate analyses in the manuscript.

6. Patients with MDR bacteremia had a higher mortality rate and the possibility of dying; could you comment on this?

7. Was there any ethical approval for this study? If so, please refer to it in the methods section.

8. The novelty of the study should be highlighted at the beginning, as should the added knowledge it brings to the scientific community.

9. The English language would need careful revisions for both consistency and clarity.

Author Response

  1. It would greatly help the reader if there were a table that included the resistance mechanism and the available treatment in the introduction section.

Response: We are grateful for your insightful recommendations regarding our manuscript. We have incorporated the following information pertaining to the mechanism of resistance and available treatment options in the introduction section;

“The most common mechanism for the development of multidrug resistance in gram-negative bacteria involved the horizontal transfer of plasmids carrying resistance genes, particularly carbapenemases [11].” (Lines 41–44)

“The utilization of newly developed agents such as tigecycline, ceftolozane-tazobactam, and ceftazidime-avibactam as empirical therapy demonstrates the superior efficacy of these agents against MDR gram-negative bacteria and thus represents a hopeful alter-native to extant therapeutics; however, the available treatment options remain inadequate [12].” (Lines 44–48)

  1. A few sentences on the epidemiology of multidrug resistance would benefit the manuscript, and general methods for antibiotic resistance would be valuable.

Response: We extend our gratitude to the reviewer for the valuable comment. We have incorporated the following information pertaining to the mechanism of resistance and available treatment options in the introduction section;

“The spread of multidrug-resistant (MDR) gram-negative bacterial isolates, including extended-spectrum β-lactamases, carbapenem-resistant Enterobacterales, Acinetobacter baumannii, and Pseudomonas aeruginosa, are on the rise worldwide [7–10].” (Lines 39–41)

“The utilization of newly developed agents such as tigecycline, ceftolozane-tazobactam, and ceftazidime-avibactam as empirical therapy demonstrates the superior efficacy of these agents against MDR gram-negative bacteria and thus represents a hopeful alter-native to extant therapeutics; however, the available treatment options remain inadequate [12].” (Lines 44–48)

  1. Clinical characteristics and outcome of Enterococcus: authors should expand on this a bit.

Response: We extend our gratitude to the reviewer for the valuable comment. The following information was included in the introduction to accentuate that Enterobacter spp. are equally challenging to treat as other MDR gram-negative bacteria;

“MDR bacterial infections, including those with MDR Enterobacterales such as Enterobacter cloacae, tend to be refractory due to the limited treatment options involving antimicrobial agents [24].” (Lines 192–194)

  1. Can authors include the hospitalization time, if available, in Table 1?

Response: We express our gratitude to the reviewer for their feedback. We have incorporated the duration of hospital stay under the subheading of hospital stays in Table 1, does this align with the reviewer's expectations?

  1. Authors should consider the association of the variables with mortality via the univariate and multivariate analyses in the manuscript.

Response: We extend our appreciation for your insightful recommendations pertaining to our manuscript. Determining the causative factors of mortality is of paramount importance. In light of univariate findings, we conducted a multivariate examination of said factors. Despite the absence of statistically significant variations, the techniques and outcomes have been incorporated into the manuscript.

“Multivariate logistic regression analysis revealed that none of the variables differed significantly as independent risk factors for early (30 days), late (30–90 days), and 90-day mortality.” (Lines 102–104)

“We performed a multivariate logistic regression analysis to identify risk factors for early (30 days), late (30–90 days), and 90-day mortality. All variables with p values of less than 0.1 in the univariable analysis were included in the multivariate analysis.” (Lines 289–291)

  1. Patients with MDR bacteremia had a higher mortality rate and the possibility of dying; could you comment on this?

Response: We are grateful for your suggestion. The augmented mortality rate associated with MDR bacteremia is deemed clinically crucial, thus we have incorporated the following information in the introduction section;

“Additionally, infections caused by MDR strains, including bacteremia, lead to increased morbidity and mortality and prolonged hospital stays and thus place a significant burden on the healthcare system [13,14].” (Lines 48–50)

  1. Was there any ethical approval for this study? If so, please refer to it in the methods section.

Response: We are grateful for your insightful recommendations regarding our manuscript. We deem it imperative to provide explicit statements regarding ethical approval, therefore we have included the following information pertaining to ethical clearance in the study design and setting in the materials and methods section;

“This study was approved by the Human Ethical and Clinical Trial Committee of Tohoku University Hospital (approval: 2019-1-270).” (Lines 265–267)

  1. The novelty of the study should be highlighted at the beginning, as should the added knowledge it brings to the scientific community.

Response: We extend our gratitude to the reviewer for the valuable suggestion. As the reviewer recommended, we have accentuated the innovativeness at the opening of the manuscript;

“Therefore, in this study, we investigated for the first time the clinical outcomes and mortality rates associated with PB due to Enterobacterales and NF-GNRs, with a focus on AMR and PB clearance.” (Lines 55–57)

  1. The English language would need careful revisions for both consistency and clarity.

Response: We express our gratitude for your suggestion. We have availed the services of an editing firm to revise the entire manuscript in English once more. Please find an attached certificate of proofreading in the English language.

Reviewer 2 Report

Article ``Impact of persistent multidrug-resistant gram-negative bacteremia on clinical outcome and mortality'' is excellently written. content interesting to the readership. The goals of the work are clearly defined, methodologically correctly done. The results of the work are presented in tables and graphics, which are adequately accompanied by discussion and appropriate references

Author Response

Article ``Impact of persistent multidrug-resistant gram-negative bacteremia on clinical outcome and mortality'' is excellently written. content interesting to the readership. The goals of the work are clearly defined, methodologically correctly done. The results of the work are presented in tables and graphics, which are adequately accompanied by discussion and appropriate references.

Response: We extend our heartfelt appreciation for your commendations. Encouraged by your positive feedback, we will persist in our research endeavors with the objective of making our findings applicative in real-world clinical settings.

Reviewer 3 Report

The topic of the manuscript is interesting and fits well the scope of the journal. However, it is a single center study with very limited sample size. The limitations of the study are obvious. Th reviewer has no objection to accept this manuscript if the authors can address these issues:

(1) What is their future study?

(2) They must define antibiotic resistance in term of MIC.

(3) The race of the patients should be indicated clearly. 

Author Response

The topic of the manuscript is interesting and fits well the scope of the journal. However, it is a single center study with very limited sample size. The limitations of the study are obvious. Th reviewer has no objection to accept this manuscript if the authors can address these issues:

(1) What is their future study?

Response: Our ultimate objective is to ascertain the characteristics of persistent bacteremia not only in GNR but also in GPC and Candida spp. and to apply these findings in real-world clinical settings. We are currently continuing our research endeavors. I will exert diligent efforts in my research to provide you with practical results in a timely manner.

(2) They must define antibiotic resistance in term of MIC.

Response: We are grateful for your insightful recommendations regarding our manuscript. We have incorporated the following definition of MIC under the subheading of definitions and outcomes in the materials and methods section, citing appropriate reference;

“The minimum inhibitory concentrations and clinical breakpoints of each bacterium were evaluated according to the Clinical Laboratory Standards Institute guidelines [32].” (Lines 277–278)

(3) The race of the patients should be indicated clearly.

Response: We are grateful for your insightful recommendations regarding our manuscript. We deemed it crucial to specify the ethnicity of the subjects in this study, therefore we have incorporated the following information pertaining to the demographic characteristics of study population in the study design and setting section of the materials and methods section and table 1 and 2.